# Essential Oil and Hydrosol Composition of Immortelle (*Helichrysum*
*italicum*)

**DOI:** 10.3390/plants11192573

**Published:** 2022-09-29

**Authors:** Nina Kunc, Antonela Frlan, Dea Baričevič, Nina Kočevar Glavač, Meta Kokalj Ladan

**Affiliations:** 1Department of Agronomy, Biotechnical Faculty, University of Ljubljana, Jamnikarjeva 101, 1000 Ljubljana, Slovenia; 2Department of Pharmaceutical Biology, Faculty of Pharmacy, University of Ljubljana, Askerceva 7, 1000 Ljubljana, Slovenia

**Keywords:** essential oil, GS–MS, *Helichrysum italicum*, hydrosol, hydrodistillation, immortelle, stems

## Abstract

The chemical composition of essential oils and hydrosols of immortelle (*Helichrysum italicum*) stems with leaves obtained by hydrodistillation was identified using gas chromatography coupled with mass spectrometry (GC-MS). One-year-old and two-year-old plants of the same immortelle population, and plants from three Croatian populations, all grown in Slovenia, were included in the study. The main compounds of essential oils of one-year-old and two-year-old plants were α-pinene, α-eudesmol, and rosifoliol. Among essential oils of the Croatian populations, three dominant components were found: α-pinene, geranyl acetate, and 2-phenylethyl tiglate. Both the essential oils and hydrosols of one-year-old plants were more diverse in their compositions than two-year-old plants. The predominant compounds of hydrosols of one- and two-year-old plants were pentan-3-one, 3-octanone, 2,2-dimethylnon-5-en-3-one, and α-terpineol, and in the Croatian populations α-terpineol, 2,6-octadien-1-ol, 2,2-dimethylnon-5-en-3-one, and α-terpineol.

## 1. Introduction

Due to an increasingly stressful lifestyle and unhealthy diet, more and more people are facing obesity, diabetes, intestinal diseases, and other chronic diseases. The solution to limiting the mentioned problems is a greater intake of plant preparations with physiologically active compounds that act against various metabolic disorders. Epidemiological studies show that regular consumption of these compounds is necessary to protect the body against chronic and infectious diseases [1]. Pharmacological and clinical tests and the experience of traditional medicine show that essential oils have the following effects when used externally: hyperemic, anti-inflammatory, antiseptic, insecticidal, and repellent. When taken orally, effects such as expectorant, appetite stimulant, choleretic, cholekinetic, carminative, antispasmodic, anti-inflammatory, antiseptic, diuretic, sedative, and circulatory stimulant effects have been observed [2].

Immortelle (*Helichrysum italicum* (Roth) G. Don) is a perennial of the family Asteraceae. The name comes from the Greek words helios (sun) and chryos (gold), referring to the typical bright yellow inflorescences that provide raw materials for the pharmaceutical, food, and cosmetic industries. The genus *Helichrysum* numbers more than 600 species, which are widespread throughout the world. Immortelle has three subspecies: *microphyllum*, *serotinum* and *italicum*. Almost 25 autochthonous species thrive in the Mediterranean area, of which *H. italicum* is the most widespread [3].

Immortelle is an aromatic shrub, 50 to 70 cm tall. The plant is a xerophyte that thrives well over a wide range of altitudes, ranging from sea level to 2200 m above sea level. It is widespread from the lower Meso-Mediterranean to lower subhumid bioclimatic environments, where it grows on calcareous and well-drained soils, preferably on sandy or clay soils. The success of immortelle cultivation depends on agroecological conditions and agrotechnical measures. Yields vary, from 8 to 20 t/ha, depending on plant spacing, irrigation, and harvest time [4,5,6]. 

Dried flowers of immortelle have been known for their healing properties since ancient times. In traditional medicine, immortelle was used as a choleretic, diuretic and expectorant. It has been found to have anti-inflammatory, antioxidant, antimicrobial, antiviral and anti-HIV properties [7]. In the last decade, interest in this plant has been growing, especially in the cosmetics and pharmaceutical industries. Its essential oil, hydrosol, as well as dried flowers and leaves, are also used in the preparation of beverages and in cooking [8]. Kramberger et al. [9], evaluated the immortelle from the point of view of effectiveness and safety for internal use by reviewing the literature. They concluded that consumption of immortelle does not pose a health risk [9].

Immortelle essential oil is valued for its pleasant smell in perfumery and cosmetic products. In vitro anti-collagenase and anti-elastase activity of the essential oil further supports the use of the plant in the cosmetic industry. Due to its wide biological potential, it also has promising applications in medicine, pharmacy, food industry and in the development of new insecticides [10]. 

Immortelle essential oil has been the subject of numerous studies, in which a large variability in chemical composition has been identified. It is important to emphasize that the chemical composition of essential oil and the content of individual ingredients depend on the plant part to be distilled, geographical origin and the growing climate, genotype of the plant species, as well as on the vegetation period of the harvest time [11].

Mastelić et al. [11] studied the yield and composition of immortelle essential oil. They used fresh plant material (stems with leaves and flower heads), which they collected near Split (south Croatia) during the flowering season. They identified 44 compounds. The main components were α-pinene (12.8%), 2-methyl-cyclohexyl pentanoate (11.1%), neryl acetate (10.4%), and 1,7-di-epi-α-cedrene (6.8%). For comparison, the research group of the same author [11] studied the content and composition of immortelle essential oil in 2005, in the flowering phase, collected at the same location as in 2008. They identified 52 substances, representing 90% of the total oil. The major identified monoterpene and sesquiterpene hydrocarbons were: α-pinene (10.2–20.6%), α-cedrene (9.6–20.5%), aromadendrene (4.4–9.4%), β-caryophyllene (4.2–8.9%), limonene (3.8–8.1%), 2,3,4,7,8,8α-hexahydro-1H-3a, 7-methanoazulene (3.0–6.3%), and ar-curcumene (2.3–4.9%). The most abundant oxygenated compounds were neryl acetate (11.5–23.2%), 2-methylcyclohexylpentanoate (8.3–12.9%), geranyl acetate (4.7–9.7%), and 2-methylcyclohexyloctanoate (4.8–7.9%). Djihane et al. [7] studied the composition of aerial parts of immortelle, which were collected during the flowering period from the north of Algeria. Analysis revealed 67 constituents representing 99.24% of the total oil. The major constituents of the oil were α-cedrene (13.61%), α-curcumene (11.41%), geranyl acetate (10.05%), limonene (6.07%), nerol (5.04%), neryl acetate (4.91%), and α-pinene (3.78%).

Most studies [3,4,11,12,13,14] have addressed the composition of essential oil obtained from flowering herbs (stems and leaves together with flowers), but only a few of them the essential oil obtained from stems and leaves (without flowers). In research conducted by Angioni et al. [1], the essential oil of stems and leaves, harvested in three different vegetation periods (from June to September), was studied. At the first harvest, in June, the major components were linalool (14.9%), neryl acetate (26.6%), and neryl propionate (14.1%). The stems, compared with flowers, were richer in limonene (17.8% compared to 5.9% in flowers) and in rosifoliol (12.1% in stems and 6.2% in flowers). The content of neryl acetate was higher in flowers (26.6%; in the stems it was 21.5%). At the second harvest, in July, the concentration of linalool was still higher in the flowers. Nerol and neryl acetate showed an opposite tendency, the former being more concentrated in the stems, the same as rosifoliol. At the third harvest, in September, the flowers were still richer in linalool, whereas the stems were richer than the flowers in neryl acetate (23.9% in the stems and 16.6% in the flowers). Concentrations of other compounds in the flowers and stems varied less.

Ferraz et al. [15] identified the following compounds in the hydrolsol: α-terpineol, 1,8-cineole, endo-borneol, terpinene-4-ol, δ-terpineol, α-terpineol, and carvacrol. Angioni et al. [12] listed neryl acetate and neryl propinoate as the main components of hydrosol from Sardinia. To the best of our knowledge, no other studies are available.

The purpose of our study was to characterize the essential oil and hydrosol of immortelle grown in Slovenia, obtained by hydro-distillation of stems with leaves (without flowers). We compared the compositions of essential oil and hydrosol obtained from one- and two-year-old seedlings of a population maintained in Slovenia with those from three Croatian populations. We performed a systematic review and comparison of the obtained results with findings in previously published scientific articles. We assume that the content and composition of essential oils and hydrosol depend on the soil, other environmental factors, and, above all, genetic factors.

## 2. Results

The chemical composition of immortelle essential oil from one- and two-year-old plants, determined by GC–MS analyses, is shown in Table 1. Essential oil components occupying at least 1% of total relative peak areas (%) in at least one of the analyzed plant groups are shown.

Essential oils of one- and two-year-old plants contained 19 and 17 compounds, respectively. In one-year-old plants, the following compounds predominated: α-pinene (19.16%), α-eudesmol (14.25%), β-eudesmol (10.79%), rosifinol (10.32%), hinesol (8.35%), guaiol (5.53%), and γ-eudesmol (5.51%). β-eudesmol (10.79%) and hinesol (8.35%) were only found in one-year-old plants. Predominant components of two-year-old plants were α-eudesmol (19.54%), rosifoliol (15.28%), α-pinene (9.4%), guaiol (9.4%), geranyl acetate (6.19%), γ-eudesmol (5.98%), and italicene (5.20%). Essential oil compounds that were only found in two-year-old plants were γ-curcumene, nerol, and valerianol, while α-pinene, rosifoliol, α-eudesmol, and γ-eudesmol were in the highest concentrations in both one- and two-year-old plants. 

The content of monoterpenes was higher in one-year-old plants. Relative peak area amounted to 32.19%. For two-year-old plants, this value was 18.9%. The content of sesquiterpenes was higher in two-year-old plants (69.37%) than in one-year-old plants (59.66%).

The total of compounds with a relative peak area of at least 1% was statistically significant (*p* = 0.03) between one- and two-year-old plants, which contained 95.49% and 92.39% of essential oil, respectively. 

In the MAP02689 population, there were 25 identified compounds, in the MAP02688 population 18 compounds, and in the MAP02685 population 22 compounds. The MAP02685 population had the highest content of α-pinene (16.19%), followed by geranyl acetate (18.43%), 2-phenyl tiglate (16.08%), and nerol (6.11%). In the MAP02688 population, the predominant compounds were α-pinene (17.75%), 2-phenyl tiglate (17.21%), geranyl acetate (9.71%), and nerol (4.9%). In the MAP02689 population, there was the highest content of α-pinene (13.5%), 2-phenyl tiglate (12.84%), geranyl acetate (5.88%), and 2,4,6,9-tetramethyldec-8-en-3,5-dion (5.4%).

The MAP02685 population had the highest content of monoterpenes (48.84%) and had the largest total relative peak area of compounds higher than 1%. The content of monoterpenes in the MAP02689 population was 26.11%, and in MAP02688 it was 35.45%. The MAP02689 population was the only one of the populations in which the content of monoterpenes was lower than the content of sesquiterpenes (43.17%). The sesquiterpene contents were 21.36% for MAP02688 and 21.26% for MAP02685.

Considering the total relative peak area of compounds higher than 1%, a significant difference among the three Croatian populations was found. Between the MAP02688 and MAP02689 populations, the *p*-value was 0.014; between MAP02685 and MAP02689, the *p*-value was 0.001; and between MAP0265 and MAP0268, the *p*-value was also 0.001. The highest value was found in the MAP02685 population (95.86%), followed by the MAP02689 population with 85.31%, and the last was the MAP02688 population with 81.28%.

Of all the detected compounds, only α-pinene, geranyl acetate, and α-curcumene were present in all samples. The compounds that were found in only one sample were p-cymene, β-curcumene, eucalyptol, 3-methyl propine, valerianol, β-eudesmol, α-acorenol, hinesol, agarospirol, and an unknown compound (ret. time 41.5) (Table 1). 

The analysis of the proportion of essential oil showed that two-year-old plants from Žalec contain a larger amount of essential oil, namely, 0.07% (0.7 mL/kg), than one-year-old plants from Žalec, which contain 0.04% (0.4 mL/kg) of essential oil. Of the populations, however, the MAP02685 population had a higher amount of essential oil (0.53% (5.3 mL/kg)) than the MAP02688 population (0.45% (4.5 mL/kg)). The MAP02685 population had the lowest amount of essential oil at 0.26% (2.6 mL/kg).

Hydrosol compounds in one- and two-year-old plants, occupying at least 1% of the total relative peak area in at least one of the analyzed plant groups, are shown in Table 2. There were 19 compounds detected in hydrosol of one-year-old plants and 21 in two-year-old plants. In one-year-old plants, the most abundant compounds were as follows: 2,2-dimethylnon-5-en-3-one (13.06%), 3-octanone (12.27%), pentan-3-one (10.06%), α-terpineol (9.14%), and 3-cyclopentene-1-ethanol (8.62%). Four of the predominant components of hydrosol occurred in both analyzed samples, but in one-year-old plants, these concentrations were higher. The most abundant compounds in two-year-old plants were 2,2-dimethylnon-5-en-3-one (8.89%), 3-octanone (7.99%), α-terpineol (7.65%), 2,6-octadien-1-ol (7.43%), and pentan-3-one (7.43%).

The content of monoterpenes was higher in one-year-old plants (14.74%) than in two-year-old plants (13.35%). The opposite was true with the sesquiterpenes content, which was higher in two-year-old plants. It amounted to 10.93% and 1.45% for one-year-old plants.

The total of compounds with a relative peak area of at least 1% of hydrosol of one-year-old plants (81.40%) was significantly higher than that of two-year-old plants (72.43%); *p* = 0.045.

The most abundant compounds in all Croatian populations were 2,2-dimethylnon-5-en-3-one (14.69–18.40%) and α-terpineol (11.8–19.55%). In the MAP02685 population, there was the highest content of 2,6-octadien-1-ol (19.09%), 2,2-dimethylnon-5-en-3-one (18.4%), α-terpineol (13.46%), linalool (5.16%), and undecane (4.17%). Predominant compounds in MAP02688 were α-terpineol (19.55%), 2,2-dimethylnon-5-en-3-one (16.29%), 2,6-octadien-1-ol (14.85%), linalol (9.54%), and 3,6-dimethyl-decane (3.21%). In the MAP02689 population these were 2,2-dimethylnon-5-en-3-one (14.69%), α-terpineol (11.80%), 2,6-octadien-1-ol (10.41%), dec-8-en-3,5-dione (9.30%), and borneol (5.14%).

The highest content of monoterpenes was detected in MAP02688 (36.89%), followed by the MAP02685 population with 29.19% and then MAP02689 with 26.31% of monoterpenes. In all three populations, only one sesquiterpene was present, namely, pogostol. Its relative peak areas ranged from 0.95% to 1.45%.

The total of compounds with a relative peak area of at least 1% in the hydrosol was the highest in the MAP02685 population (94.7%). The value in the MAP02688 population was 93.58%, and the value was lowest in the MAP02689 population (81.28%). There were statistically significant differences between MAP02688 and MAP02689 populations (*p* = 0.001) and between MAP02685 and MAP02689 populations (*p* = 0.001). There were no statistically significant differences between the MAP02685 and MAP02688 populations (*p* = 0.58). 

Examining all the compounds, it can be concluded that nine compounds are common to all hydrosols (Table 2). These are pentan-3-one, isopropyl ethyl ketone, 1-(1,1-dimethylethoxy)-2,2-dimethyl-propane, 3-octanone, 3,6-dimethyl-decane, 2,6-octadien-1-ol, borneol, 2,2-dimethylnon-5-en-3-one, and α-terpineol. The compounds that were found in only one sample were verbenol hexanol, hex-(3Z)-enol, 2,5-dimethylfuran-3.4(2H, 5H)-dione, sabinene hydrate, guaiol, α-eudesmol, γ-eudesmol, allyl 2-ethyl butyrate, nerol, and 3-cyclopentene-1-ethanol.

## 3. Discussion

When comparing results from different scientific studies, it is necessary to consider that in our study, immortelle was grown in heavy clay soils, which probably affected the composition of essential oils. It is well known that soil type, as well as other ecological conditions and the developmental stage affect the chemical composition of plant material [16].

We found that there were differences in essential oils of one- and two-year-old plants. Two-year-old plants contained more α-eudesmol, guaiol, geranyl acetate, rosifoliol, italicene, and γ-eudesmol than one-year-old plants, but less α-pinene and bulnesol. In contrast to this, one-year-old plants contained hinesol, α-acorenol, and agarospirol, which were not detected in two-year-old plants. In addition, two-year-old plants contained γ-cucumene, nerol, and valerianol, which were not detected in one-year-old plants. We conclude that one-year-old plants contained more monoterpenes than did two-year-old plants, and less sesquiterpenes.

A high value of α-pinene (24.58%) was found in the developmental stages of early stems, and a low value in the flowering phase and after flowering [11]. In our research, in which we distilled stems with leaves and without flowers, α-pinene was present in all analyzed samples. The closest mentioned values were in essential oils of one-year-old plants (19.16%), followed by the MAP02688 (17.75%), MAP02685 (16.19%), and MAP02689 (13.5%) populations, and the lowest value for two-year-old plants (9.04%). Morone-Fortunato et al. [14] reported that α-curcumene was low (0.44%) in early stems and high in the flowering phase (28%). In our samples, the value of α-curcumene was also low. It was 0.56% for essential oil of one-year-old plants and 1.49% for two-year-old plants, and between 2.1% and 2.79% for Croatian populations. Morone-Fortunato et al. [14] also reported γ-curcumene content, which was highest in the developmental phase of early stems with leaves (16.65%), and in the flowering phase it decreased to 12.03%. In our case, its content in the Croatian populations ranged from 1.46% to 3.46%, and in two-year-old plants it was 3.12%. However, γ-curcumene was not present in one-year-old plants. 

Ninčević et al. [3], in reviewing the literature, summarized the composition of the essential oils of immortelle populations that thrive in the former Yugoslavia. They found that the essential oils of Croatian immortelle were characterized by a high content of α-pinene (22%), followed by y-curcumene (10%), β-selinene (6%), neryl acetate (6%), and β-caryophyllene (5%), while along the Adriatic Coast, the main compounds were α-curcumene (15–29%) or γ-curcumene or α-pinene (25–30%) and neryl acetate (4–14%). Our MAP02689 population, which we received from Croatia (Cavtat), also had the highest content of α-pinene (13.5%) but a lower content than that reported by Ninčević et al. [3]. In addition, our MAP02689 and MAP02685 populations had a high content of 2-phenylethyl tiglate, which was not reported in any other literature.

Ninčević et al. [3] reported that the content of neryl acetate was highest in the stage of development of early stems (9.02%) and lowest after flowering (5.57%), which indicates that neryl acetate is the highest in the youngest parts of the plant (stems) and lower in later stages (after flowering). Neryl acetate was not present in our samples. In contrast to this, geranyl acetate was present in our samples (6.19% in two-years-old plants, 2.94% in one-year plants, and from 5.88% to 18.43% in Croatian populations). The relatively high percentage (18.43%) of geranyl acetate meant that it was the compound with the highest share in the MAP02685 population. It can be assumed that monoterpenes were predominant in the early stages of plant growth and development, while sesquiterpenes predominated during flowering and after flowering.

Analysis of the chemical composition of the Croatian populations showed that the predominant compounds found in their essential oils were α-pinene, 2-phenylethyl tiglate, and geranyl acetate. All three compounds were lowest in the MAP02689 population. However, this population contained more α-copaene, 2,4,6,9-tetramethyldec-8-ene-3,5-dione (not present in MAP02688), α-cadinol, as well as linalool (not present in MAP02688). Β-curcumene and italicene were found only in the MAP02689 population. The content of α-pinene was highest in MAP02688 (17.75%), and that of geranyl acetate and of nerol (not present in MAP02689) were highest in the MAP02685 population (18.43% and 6.11%, respectively).

Blažević et al. [16] studied the composition of immortelle essential oil in nine natural habitats in Croatia. Forty-four compounds were identified, the main ones of which were α-pinene, neryl acetate, α-cedar, nerol, α-curcumene, γ-curcumene, and geranyl acetate. The composition of oil was similar in all nine habitats but differed in the number of individual compounds. Their results are most similar to the results of our study. 

Oji and Shafaghat [17] studied the composition of *H. armenium* essential oil in flowers, leaves, and stems of plants from Iran. The predominant compound, both in leaves and stems, was limonene, which is again different from our research and from the research we have already mentioned. In second place was α-pinene, the content of which was 14.4% in leaves and 13.4% in stems. In the Croatian MAP02689 population, the value of α-pinene was also among the highest. This represented 15.53%, which is slightly higher than in the study by Oji and Shafaghat [17]. They reported high values (11.9%) of borneol in flowers, and very low values (2.3 to 2.6%) in other parts of the plant. The presence of α-gurjunena in leaves and stems (6.3%) was also determined, but it was not detected in flowers. We did not identify this compound and did not find any other study reporting α-gurjunenes. However, our results agree with Oji and Shafaghat [17] in the content of δ-cadinen, which was 1.47 to 1.6% in our populations, while the mentioned authors stated values of 1.5 to 2.2%. Both also identified the presence of α-cadinol, which was higher in our MAP02689 and MAP02688 populations (3.1–4.29%) than the values (1.5–2.2%) reported by Oji and Shafaghat [17]. Their content was similar to the content of the mentioned compound in our MAP02685 population (1.92%). According to Oji and Shafaghat [17], the content of α-cadinol was extremely high in flowers (18.2%).

Looking at the scarce research about the composition and content of immortelle hydrosols, it can be said that it is extremely diverse. Comparing the results of our research with the results reported by Ferraz et al. [9], who analyzed plants from Portugal, monoterpenes also predominated in samples obtained from plants grown in Slovenia. The main compound of Portuguese hydrosols was α-terpineol, with 30.5%, followed by carvacrol with 29.6% and cineole with 1.8%. Other components ranged from 2 to 6.6%. For our samples of one- and two-year-old plants from this study, these values were different; 2,2-dimethylnon-5-en-3-one was predominant in one-year-old plants (13.06%) and in two-year-old plants (8.89%). It can be seen that our compounds with the highest proportion were more than half as low. The number of compounds we detected was significantly higher than stated by the mentioned authors, which indicates a higher diversity of our samples. Slightly higher values were obtained in the Croatian populations, in which the compound 2,2-dimethylnon-5-en-3-one (18.4%) was present in the highest proportion in the MAP02685 population, as well as in two-year-old plants. The MAP02688 population was dominated by α-terpineol, which is the same as that reported by Ferraz et al. [15], but this share was slightly lower in the Slovene samples at 19.55%. In the MAP02689 population, it accounted for 11.8%. Angioni et al. [12] listed neryl acetate (23.9%) as the main component of immortelle hydrosols from Sardinia, harvested in September, followed by neryl propinoate (13.0%), which were not identified in our samples. Unlike that reported by Ferraz et al. [15], the α-terpineol content was only 9.5%, which is lower than the values obtained in our study. The fourth most represented component was γ-curcumene (9.3%), which was not detected in our study, nor was it identified by Ferraz et al. [15].

Comparing the compounds that occupy at least 3% of the essential oils or the hydrosols, the compounds present in both the essential oils and hydrosols are the following: geranly acetate, guaiol, rosifoliol, α-eudesmol, γ-eudesmol, and nerol. The other compounds did not occur in either hydrosols or essential oils, which means that they are not distributed in the aqueous phase of the hydrosol due to the higher logP value (Table 3). The value of the partition coefficient (logP) of an individual compound plays a decisive role in the occurrence or, more precisely, in the content itself. Compounds with lower logP are better distributed in the aqueous phase and those with higher logP in the lipophilic phase. After reviewing the logP values of all compounds appearing in the hydrosol of the first fraction of fresh or dried herbal drug, we found that no compound with a logP greater than 3.9 appeared in the hydrosols.

## 4. Materials and Methods

### 4.1. Plant Material

Samples of one- and two-year-old immortelle plants and samples of plants grown from seeds from three different Croatian populations were used in the research work. One- and two-year-old immortelle plants were obtained from the Institute of Hop Growing and Brewing of Slovenia in Žalec (46.24979 N, 15.16274 E, 255 m), where a population of immortelle is maintained and multiplied every year. Seeds from Croatian populations of immortelle (MAP02685, MAP02688, and MAP02689) were obtained from the Scientific Center for Biodiversity and Molecular Breeding of Plants in Zagreb, Croatia. The natural locations from which the seeds were obtained are shown in Table 4.

The obtained seeds were sown in a greenhouse of the Biotechnical Faculty, University of Ljubljana, in plant germinators, and the seedlings were then transplanted to the field of the Biotechnical Faculty. A total of 497 plants was planted. We did not use any fertilizers in the experiment, not even plant protection products. We used only mechanical tillage methods, such as hand tillage.

Harvesting of the aboveground part (stems with leaves, without flowers, because plants were in a vegetative developmental stage) of the immortelle was carried out in October 2020. Leaves and stems were collected, with 4 replications performed in each population. The plant material was dried for one week at room temperature and then distilled. 

### 4.2. Hydrodistillation

The essential oil was isolated by hydrodistillation according to the procedures of the European Pharmacopoeia [19] using a Clevenger apparatus. According to the method described by Anžlovar et al. [20], the distillation of 200 g of dry plant mass took 2 h. No solvents were used in the distillation process. After distillation, the essential oil and hydrosol were collected in dark vials and stored in a refrigerator at 4 °C until further analysis on a gas chromatograph, together with a mass spectrometry detector.

### 4.3. GC–MS Analyses

Gas chromatography with a mass spectrometry detector (GC–MS) was performed with a QP-2010 Ultra (Shimadzu, Kyoto, Japan), Rxi-5Sil MS column, 30 m × 0.25 mm i.d., film thickness 0.25 µm (Restek, Bellefonte, PA, USA) under the following conditions: temperature program 40 °C, 3 °C/min to 220 °C, 220 °C (15 min). Injector temperature 250 °C; ion source temperature 200 °C; interface temperature 300 °C; injection volume 1 µL; division 1:100; carrier gas He; carrier gas flow 1 mL/min. The conditions of mass spectrometry were electron impact mode, total ionic current record, 1 kV detector voltage. The mass spectra of the compounds obtained were compared with the spectra from the NIST14 (National Institute of Standards and Technology, Gaithersburg, MD, USA) and FFNSC 3 (Shimadzu, Kyoto, Japan) mass spectrum libraries. Concentrations were calculated as relative peak areas in %. Hydrosols were injected directly, without prior extraction, essential oils were diluted in hexane, and the concentration was 1% vol/vol.

### 4.4. Statistical Analysis

The results were analyzed with the R commander statistical program using one-way analysis of variance (ANOVA). The Duncan test was used in the case of the content of essential oil from one- and two-year-old plants, to compare treatments, when ANOVA showed significant differences among values. In this case, we used this test because we only had two treatments (one- and two-year-old plants). The Tukey HSD test was used in the case of essential oil content from Croatian populations and between populations and one- and two-year-old plants to compare treatments when ANOVA showed significant differences among values. We used this test because we had three or more different treatments. Results are given as the mean value with standard deviation (SD). If the *p*-values were lower than 0.05, then the differences among treatments were statistically significant.

## 5. Conclusions

Knowledge of the composition of essential oil or hydrosol is important from the point of view of industrial use—ensuring the same quality of these substances as raw materials; for example, the composition has a strong influence on the smell, which is essential when used in the cosmetics industry.

The results of the study confirmed our assumptions that the content and composition of essential oils and hydrosols depend on the soil, other environmental factors, and especially genetic factors, which was further confirmed by comparing our results with the existing literature.

Essential oils of one- and two-year-old plants of the population that has been maintained and multiplied for several years by the Institute of Hop Growing and Brewing of Slovenia in Žalec and is presumed to be well adapted to the Slovene climate contained 19 and 17 compounds, respectively. In both one- and two-year-old plants of this population, α-pinene, rosifoliol, α-eudesmol, and γ-eudesmol were identified as the major compounds in the essential oils. The content of monoterpenes was higher in one-year-old plants (32.19%) than in two-year-old plants (18.9%), but sesquiterpenes were more abundant in older plants (69.37%) than in one-year-old plants (59.66%). The content of monoterpenes in hydrosols was slightly higher in one-year-old plants (14.74%) than in two-year-old plants (13.35%), while the difference in the content of sesquiterpenes was more evident in older plants.

Among all populations included in our study, the richest in essential oil composition was found to be that of the MAP02689 population when grown in the ecological conditions of central Slovenia. In the MAP02689 population, there were no significant differences between monoterpenes in essential oils and hydrosols. In the MAP02688 population, there was a slightly higher relative peak area % of monoterpenes in hydrosols (36.89%). The value of sesquiterpenes was much higher in essential oils compared to hydrosols. The MAP02685 population was characterized by the highest content of monoterpenes (48.84%) and the largest relative peak area of compounds higher than 1%. 

The results of the study also indicate that the compounds present both in hydrosols and essential oils (geranium acetate, guaiol, rosifoliol, α-eudesmol, γ-eudesmol, and nerol) have a lower logP value, which means that they are better distributed in the aqueous phase.

## Figures and Tables

**Table 1 plants-11-02573-t001:** Average relative peak area ± standard deviation (SD) (%) of individual compounds in essential oils (average ± SD) in two- and one-year-old immortelle plants (*H. italicum*) and in three Croatian populations (MAP02685, MAP02688, MAP02689) of immortelle plants *(H. italicum*). Different letters indicate statistical differences between plants.

Compound	One-Year-Old Plants	Two-Year-Old Plants	MAP02685	MAP02688	MAP02689
α-pinene (ret. time 10.525)	19.16 ± 1.72 a	9.04 ± 1.85 c	16.19 ± 1.34 b	17.75 ± 0.52 b	13.5 ± 1.05 c
p-cymene (ret. time 14.778)	-	-	-	-	1.07 ± 0.15
limonene (ret. time 15.006)	-	-	1.84 ± 0.44 b	3.09 ± 0.12 a	2.79 ± 0.18 a
isobutyl angelate (ret. time 15.918)	-	-	0.83 ± 0.008 a	-	1.06 ± 0.24 a
isoamyl angelate (ret. time 20.795)	-	-	2.12 ± 0.18 ab	1.64 ± 0.23 a	2.29 ± 0.28 b
geranyl acetate (ret. time 30.385)	2.94 ± 1.17 d	6.19 ± 1.58 c	18.43 ± 0.99 a	9.71 ± 0.36 b	5.88 ± 1.54 bc
α-copaene (ret. time 31.194)	-	-	-	2.12 ± 0.49 a	2.38 ± 0.95 a
sesquitujen (ret. time 32.815)	-	-	2.09 ± 0.48 a	2.05 ± 0.24 a	2.57 ± 0.82 a
8-decene-3,5-dione, 2,4,6,9-tetramethyl (ret. time 35.342)	-	-	2.1 ± 0.26 a	2.6 ± 0.27 a	1.89 ± 0.87 a
2,4,6,9-tetramethyldec-8-en-3,5-dion (ret. time 35.517)	-	-	2.61 ± 0.26 b	-	5.4 ± 0.9 a
α-curcumene (ret. time 35.641)	0.56 ± 0.26 d	1.49 ± 0.16 c	2.79 ± 0.33 a	2.1 ± 0.22 b	2.38 ± 0.45 b
β-curcumene (ret. time 35.489)	-	-	-	-	2.6 ± 1.42
γ-curcumene (ret. time 35.390)	-	3.12 ± 0.48 a	3.64 ± 0.51 a	2.24 ± 0.24 a	1.46 ± 0.57 a
δ-cadinene (ret. time 37.188)	-	-	-	1.6 ± 0.14 a	1.47 ± 0.43 a
2-phenylethyl tiglate (ret. time 37.811)	-	-	16.08 ± 0.29 a	17.21 ± 0.2 a	12.84 ± 0.95 b
rosifoliol (ret. time 40.913)	10.32 ± 1.90 b	15.28 ± 1.42 a	-	2.42 ± 0.28 c	1.4 ± 0.57 d
neointermedeol (ret. time 41.106)	-	-	1.13 ± 0.15 a	1.67 ± 0.4 a	1.86 ± 0.94 a
l-α-cadinol (ret. time 42.523)	-	-	1.92 ± 0.93 c	3.1 ± 0.21 b	4.29 ± 0.37 a
pogostol (ret. time 42.643)	-	-	-	2.7 ± 0.23 a	1.72 ± 0.97 b
e-caryophyllene (ret. time 33.084)	-	-	2.97 ± 0.23 b	-	2.03 ± 0.24 a
isoitalicene (ret. time 32.475)	-	-	2.08 ± 0.25 a	-	2.63 ± 0.74 a
β-accordien (ret. time 35.503)	1.02 ± 0.33 a	1.25 ± 0.15 ab	2.86 ± 0.29 b	1.36 ± 0.085 a	-
neryl propionate (ret. time 34.234)	1.74 ± 0.72 b	2.61 ± 0.57 a	1.1 ± 0.39 c	-	-
eucalyptol (ret. time 15.105)	-	-	1.71 ± 0.62	-	-
α-cedrene (ret. time 31.116)	1.05 ± 0.08 c	0.69 ± 0.16 d	1.78 ± 0.43 b	-	2.21 ± 0.61 a
nerol (ret. time 24.239)	-	1.06 ± 0.13 c	6.11 ± 0.45 a	4.9 ± 0.28 b	-
linalool (ret. time 18.351)	-	-	3.46 ± 0.55 a	-	2.87 ± 0.96 b
3-methylapopinene	-	-	2.02 ± 0.41	-	-
italicene (ret. time 32.472)	2.51 ± 1.24 b	5.20 ± 0.83 a	-	-	2.8 ± 0.19 b
hexyl angelate (ret. time 26.943)	1.22 ± 0.32 a	1.97 ± 0.73 a	-	-	-
guaiol (ret. time 40.181)	5.53 ± 2.07 b	9.4 ± 0.95 a	-	-	-
α-eudesmol (ret. time 42.416)	14.25 ± 2.05 b	19.54 ± 0.84 a	-	-	-
β-eudesmol (ret. time 42.387)	10.79 ± 2.22	-	-	-	-
γ-eudesmol (ret. time 41.633)	5.51 ± 1.21 a	5.98 ± 0.15 a	-	-	-
bulnesol (ret. time 42.887)	4.79 ± 0.35 a	3.19 ± 0.27 b	-	-	-
valerianol (ret. time 42.912)	-	4.23 ± 0.51	-	-	-
hinesol (ret. time 41.955)	8.35 ± 2.31	-	-	-	-
α-acorenol (ret. time 41.779)	2.24 ± 0.5	-	-	-	-
agarospirol (ret. time 40.279)	1.09 ± 0.25	-	-	-	-
unknown (ret. time 40.07)	1.08 ± 0.28 b	2.15 ± 0.17 a	-	-	-
unknown (ret. time 41.5)	1.34 ± 0.53	-	-	-	-
TOTAL **	95.49 ± 19.51 a	92.39 ± 10.95 ab	95.86 ± 9.79 a	81.28 ± 4.69 d	85.31 ± 17.28 c

** TOTAL (of compounds with a relative peak area of at least 1%).

**Table 2 plants-11-02573-t002:** Average relative peak area ± standard deviation (SD) (%) of individual compounds in hydrosol (average ± SD) in two- and one-year-old immortelle plants (*H. italicum*) and in three Croatian populations (MAP02685, MAP02688, MAP02689) of immortelle plants (*H. italicum*). Different letters indicate statistical differences between plants.

Compound	One-Year-Old Plants	Two-Year-Old Plants	MAP02685	MAP02688	MAP02689
pentan-3-one (ret. time 3.728)	10.06 ± 1.80 a	7.43 ± 1.53 b	3.58 ± 0.79 c	2.63 ± 0.87 c	2.99 ± 1.60 c
carbinol (ret. time 4.425)	-	-	1.50 ± 0.54 a	-	2.06 ± 1.45 a
isopropyl ethyl ketone (ret. time 4.601)	2.52 ± 0.94 a	1.79 ± 0.61 a	2.99 ± 0.48 a	1.78 ± 0.6 a	2.34 ± 1.51 a
2,2,3,3-tetramethyl pentane (ret. time 4.833)	-	-	-	2.93 ± 0.71 a	2.00 ± 1.07 a
1-(1,1-dimethylethoxy)-2,2-dimethyl-propane (ret. time 6.963)	2.38 ± 0.50 a	2.54 ± 0.94 a	2.54 ± 0.49 a	2.52 ± 0.69 a	1.64 ± 0.82 a
hex-(2E)-enal (ret. time 7.588)	1.1 ± 0.35 a	-	1.93 ± 0.55 a	1.64 ± 0.34 a	1.49 ± 0.56 a
hexanol (ret. time 8.063)	-	-	1.09 ± 0.41	-	-
hex-(3Z)-enol (ret. time 7.566)	-	-	1.02 ± 0.31	-	-
3-octanone (ret. time 11.700)	12.27 ± 0.88 a	7.99 ± 1.1 b	2.89 ± 0.96 c	2.30 ± 1.02 c	1.92 ± 0.96 c
eucalyptol (ret. time 15.040)	-	-	2.73 ± 1.46 a	2.85 ± 0.82 a	2.78 ± 1.88 a
3,6-dimethyl-decane (ret. time 17.934)	1.31 ± 0.43 c	2.38 ± 0.94 b	3.45 ± 0.89 a	3.21 ± 1.28 a	2.40 ± 1.32 a
undecane (ret. time 18.048)	3.40 ± 0.75 a	2.54 ± 0.86 b	4.17 ± 1.52 a	2.63 ± 0.93 b	-
2,6-octadien-1-ol (ret. time 24.401)	4.35 ± 2.05 d	7.43 ± 2.65 c	19.09 ± 3.91 a	14.85 ± 3.39 b	10.41 ± 4.96 b
dec-8-en-3,5-dione (ret. time 33.647)	-	-	1.98 ± 0.76 b	2.46 ± 0.97 b	9.30 ± 3.23 a
borneol (ret. time 21.907)	1.31 ± 0.69 b	1.37 ± 0.47 b	3.57 ± 1.03 b	3.04 ± 0.46 b	5.14 ± 2.38 a
carvacrol (ret. time 27.934)	-	-	1.63 ± 0.35 a	-	1.69 ± 1.07 a
2,2-dimethylnon-5-en-3-one (ret. time 22.176)	13.06 ± 2.00 b	8.89 ± 1.43 b	18.4 ± 4.97 a	16.29 ± 3.30 a	14.69 ± 2.43 a
pogostol (ret. time 42.522)	-	-	0.95 ± 0.3 a	1.30 ± 0.62 a	1.45 ± 0.88 a
α-terpineol (ret. time 22.841)	9.14 ± 0.93 c	7.65 ± 1.65 d	13.46 ± 3.87 b	19.55 ± 6.22 a	11.80 ± 2.63 c
unknown (ret. time 28.08)	1.19 ± 0.26 a	-	-	1.08 ± 0.3 a	-
8-decene-3,5-dione (ret. time 35.214)	-	-	-	1.07 ± 0.37 b	2.28 ± 0.68 a
linalool (ret. time 17.627)	2.4 ± 0.92 c	-	5.16 ± 0.89 b	9.54 ± 8.22 a	4.90 ± 1.68 b
eucalyptol (ret. time 15.137)	-	2.37 ± 0.76			
3,6-dimethyl-4-octanone (ret. time 18.000)	1.81 ± 0.82 b	3.08 ± 1.61 a	-	-	-
verbenol (ret. time 20.578)	-	1.05 ± 0.42	-	-	-
guaiol (ret. time 40.208)	-	2.18 ± 0.36	-	-	-
nerol (ret. time 30.313)	1.89 ± 0.58	-	-	-	-
geranyl acetate (ret. time 33.510)	-	2.17 ± 0.53	2.64 ± 0.97 a	1.91 ± 0.87 a	-
sabinene hydrate (ret. time 18.390)	-	1.11 ± 0.29	-	-	-
rosifoliol (ret. time 40.901)	1.45 ± 0.41 b	3.34 ± 0.76 a	-	-	-
α-eudesmol (ret. time 42.493)	-	4.09 ± 0.99	-	-	-
γ-eudesmol (ret. time 41.557)	-	1.32 ± 0.54	-	-	-
allyl 2-ethyl butyrate (ret. time 13.663)	1.93 ± 0.40	-	-	-	-
3-cyclopentene-1-ethanol (ret. time 23.287)	8.62 ± 1.31	-	-	-	-
propan-2-ol (ret. time 7.071)	-	1.76 ± 0.34	-	-	-
pentanoic acid	1.21 ± 0.31 a	1.71 ± 0.70 a	-	-	-
TOTAL **	81.40 ± 16.3	72.43 ± 15.92 b	94.7 ± 25.45 a	93.58 ± 31.98 a	81.28 ± 31.11 c

** TOTAL (of compounds with a relative peak area of at least 1%).

**Table 3 plants-11-02573-t003:** Comparison of essential oils and hydrosols of immortelle plant (*H. italicum*) content for the average amount of compounds that occurred in the essential oil in a relative peak area greater than 3%. LogP values were read from the PubChem web portal [18].

Compound	Average Essential Oil Content (%)	Average Hydrosols Content (%)	LogP
α-pinene	15.13	**-**	2.8
geranyl acetate	7.19	2.24	3.5
2-methylene-4,8,8-trimethyl-4-vinyl	3.47	-	5.1
2,4,6,9-tetramethyldec-8-en-3,5-dion	4.01	-	3.9
2-phenylethyl tiglate	15.38	-	3.3
bulnesol	3.99	-	2.9
guaiol	7.47	1.89	3.1
valerianol	4.23	-	3.5
hinesol	8.35	-	3.7
rosifoliol	7.36	2.39	3.9
l-α-cadinol	3.1	-	3.3
α-eudesmol	16.89	4.09	3.5
β-eudesmol	10.79	-	3.7
γ-eudesmol	5.75	1.93	3.4
linalool	3.17	-	2.7
italicene	3.5	-	3.4
nerol	1.99	1.89	2.9

**Table 4 plants-11-02573-t004:** Locations from which we obtained seeds of immortelle natural populations from Croatia.

Population	Location	Latitude	Longitude	Altitude (m)
MAP02685	Lošinj	44 3540 N	14 2442 E	73
MAP02688	Obrovac	44 1312 N	15 4019 E	137
MAP02689	Cavtat	42 3516 N	18 1534 E	525

## Data Availability

All data are presented within the article.

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
