# Peer review of "Essential Oil and Hydrosol Composition of Immortelle (Helichrysumitalicum)"

_plants, 2022, doi:10.3390/plants11192573_

Round 1

Reviewer 1 Report

For my field of expertise, the article is well presented, the scientific content is consistent with the references and preliminary results. The authors expand research in order to validate the scientific results. The article can be published in its current form.

Author Response

Dear Reviewer 1,

thank you for your positive comment on our article. 

Best regards,

Nina Kunc

Reviewer 2 Report

This article studied essential oil and hydrosol composition of immortelle. The benefits of immortelle and its essential oils are well known therefore this study will help in the production of the oils and hydrosols. Before recommending this article for publication, there are some shortcomings for that should be resolve.

Abstract

Abstract is well written.

Introduction

Introduction should be start from general to specific. Add few lines on significance of plant oils, medicinal and traditional importance, leading to discussion about immortelle. The following studies could be cited. Pak. J. Bot., 54(3): DOI: http://dx.doi.org/10.30848/PJB2022-3(19),

http://doi.org/10.36899/JAPS.2022.3.0484,

Add commercial and industrial importance of Immortelle.

numerous studies line 48 must be cited.

Line 72-72 most studies must be cited.

What is significance of this study. As many studies reported isolation of essential oils and hydrosols from plants.

Methods

Section 4.2. provide complete details of the paper.

The result and discussion are well presented

Conclusion is well justified. The authors should discuss some points for the future studies molecular level studies are required to know about the involved mechanism and improvement of oil productivity.

Author Response

Dear Reviewer 2,

Thank you for your review of the paper " Essential oil and hydrosol composition of immortelle (Helichrysum italicum)" and useful comments.

We have read the queries and corrected the paper according to your suggestions. We have answered the comments, which we believe have helped us to improve the reports of our study for publication in Plants. For your convenience, we've marked the changes we've made in the article and labeled them R2.

Reviewer 3 Report

These are my main comments on the manuscript (plants-1924150) entitled “Essential oil and hydrosol composition of immortelle (Helichrysum italicum)”. The manuscript investigates the chemical composition of the essential oils and hydrolates of immortelle (Helichrysum italicum) obtained by hydrodistillation and characterized by GC-MS. Following substantial revisions should be incorporated in the manuscript prior to acceptance.

1. I have concerns about the manuscript sections that I believe need to be addressed in order to improve its clarity.

2. A hypothesis for this work is needed.

3. In results section, Duncan's multiple range test is not rigorous, I suggest replace by Tukey HSD test.

4. Conclusion section should be summarized

5. Other revisions could be checked in PDF attached.

Author Response

Dear Reviewer 3,

Thank you for your review of the paper " Essential oil and hydrosol composition of immortelle (Helichrysum italicum)" and useful comments.

We have read the queries and corrected the paper according to your suggestions. We have answered the comments, which we believe have helped us to improve the reports of our study for publication in Plants. We have taken into account the changes that we think are appropriate according to our article and also according to the suggestions of the other two reviewers. For your convenience, we've marked the changes we've made in the article and labeled them R3.

Round 2

Reviewer 3 Report

The authors have incorporated all suggestions and comments into the revised version, now the manuscript seems much clear. There is minor point to be corrected:

Ls.20, 35, 42, etc: For references, change “[4],[20],[16], etc.” by “[1],[2], [3], etc.” according to the journal style. Please, correct in all manuscript.

Ls.40-42: Combine and summarize these sentences

L.42: “and” (no italic)

L.60: “In vitro” should be in italic

Author Response

Dear Reviewer 3,

Thank you for your review (Round 2) of the paper " Essential oil and hydrosol composition of immortelle (Helichrysum italicum)" and useful comments.

We have read the queries and corrected the paper according to your suggestions. We have answered the comments, which we believe have helped us to improve the reports of our study for publication in Plants. For your convenience, we've marked the changes we've made in the article and labeled them R3 (Round 2).
